# Prognostic Value of C-Reactive Protein-to-Albumin Ratio in Head and Neck Cancer: A Meta-Analysis

**DOI:** 10.3390/diagnostics11030403

**Published:** 2021-02-26

**Authors:** Chih-Wei Luan, Hsin-Yi Yang, Yao-Te Tsai, Meng-Chiao Hsieh, Hsin-Hsu Chou, Kuo-Su Chen

**Affiliations:** 1Department of Otorhinolaryngology-Head and Neck Surgery, LO-Sheng Hospital Ministry of Health and Welfare-Home, New Taipei City 242, Taiwan; jackluan2010@gmail.com; 2Graduate Institute of Clinical Medical Sciences, College of Medicine, Chang Gung University, Taoyuan 333, Taiwan; 3Faculty of Medicine, College of Medicine, Chang Gung University, Taoyuan 333, Taiwan; 4Clinical Medicine Research Center, Ditmanson Medical Foundation Chia-Yi Christian Hospital, Chiayi City 600, Taiwan; cych13018@gmail.com; 5Department of Otorhinolaryngology-Head and Neck Surgery, Chang Gung Memorial Hospital, Chiayi 613, Taiwan; yaote1215@gmail.com; 6Division of Colon and Rectal Surgery, Department of Surgery, Chang Gung Memorial Hospital Chiayi Branch, Chiayi 613, Taiwan; mr8872@gmail.com; 7Department of Pediatric, Chiayi Christian Hospital, Chiayi City 600, Taiwan; 03834@cych.org.tw; 8Department of Nephrology, Chang Gung Memorial Hospital, Keelung 20401, Taiwan

**Keywords:** C-reactive protein-to-albumin ratio, head and neck cancer, meta-analysis, overall survival, disease-free survival, distant metastasis–free survival

## Abstract

The C-reactive protein-to-albumin ratio is a proven prognostic predictor of nasopharyngeal carcinoma. However, the role of the C-reactive protein-to-albumin ratio in other head and neck cancers remains unclear. This meta-analysis explored the prognostic value of the C-reactive protein-to-albumin ratio in head and neck cancers. A systematic search was conducted. Outcomes of interest included overall survival, disease-free survival, and distant metastasis–free survival. The hazard ratio with 95% confidence interval was pooled using a random-effects model. A total of 11 publications from the literature were included, allowing for the analysis of 7080 participants. Data pooling demonstrated that pretreatment C-reactive protein-to-albumin ratio had a hazard ratio of 1.88 (95% CI: 1.49−2.37, *p* < 0.001) for predicting overall survival, 1.91 (95% CI: 1.18−3.08, *p* = 0.002) for disease-free survival, and 1.46 (95% CI: 1.08−1.96, *p* = 0.001) for distant metastasis–free survival. Subgroup analysis showed that the C-reactive protein-to-albumin ratio is a significant prognostic marker for various head and neck cancers. An elevated pretreatment C-reactive protein-to-albumin ratio predicts a worse prognosis for patients with head and neck cancers. Therefore, the C-reactive protein-to-albumin ratio could serve as a potential prognostic biomarker facilitating treatment stratification.

## 1. Introduction

Head and neck cancer (HNC), including cancers of the nasopharynx, oral cavity, oropharynx, hypopharynx, and larynx, accounts for 800,000 of all new cancer diagnoses each year and caused more than 400,000 deaths in 2018 [1]. The mainstay of treatment for HNC is either ablative surgery with or without adjuvant therapy or definite chemoradiotherapy. However, despite aggressive treatment and multidisciplinary management, the overall prognosis of HNC remains poor, and patients with the same TNM staging frequently have disparate survival outcomes [2]. Hence, an improved understanding of carcinogenesis and the identification of straightforward and readily available biomarkers of HNC may contribute to creating individualized treatment with more precise patient stratification and prognosis prediction [3,4].

Accumulated evidence has demonstrated the roles of systemic inflammation and host immunity in angiogenesis and cancer progression [5,6]. Studies have also demonstrated that many inflammation-based scores, including neutrophil-to-lymphocyte ratio, Glasgow prognostic score, lymphocyte-to-monocyte ratio, and platelet-to-lymphocyte ratio, are related to the prognosis of human malignancies [7]. The C-reactive protein-to-albumin ratio (CAR), incorporating host nutritional index and systemic inflammation status, is an independent prognostic biomarker for several cancers, such as small-cell lung cancer, esophageal squamous cell carcinoma, hepatocellular carcinoma, renal cell carcinoma, colorectal cancer, and pancreatic cancer [8,9,10,11,12,13]. CAR measurement is readily available in pretreatment blood tests and is cost effective compared with other serum biomarkers. Therefore, CAR can help clinicians predict treatment outcomes.

Two meta-analyses have reported the prognostic value of CAR for determining nasopharyngeal cancer (NPC) survival [14,15]. However, the prognostic value of CAR for HNC and other than NPCs has not been extensively reviewed. Thus, we performed this meta-analysis to answer the following research question: Is CAR a significant prognostic biomarker in all HNCs?

## 2. Materials and Methods

### 2.1. Data Sources and Search Strategy

A systemic review and meta-analysis was performed according to the Preferred Reporting Items for Systematic Reviews and Meta-Analysis criteria [16]. A structured search was conducted of the US National Library of Medicine (PubMed), the Cochrane CENTRAL Register of Controlled Trials (CENTRAL), and the Excerpta Medica database (Embase) for relevant trials from inception to 30 June 2020. The search keywords were “(C-reactive protein-to-albumin ratio OR CRP/Alb ratio OR C-reactive protein/albumin ratio) AND (nasopharyngeal OR oropharyngeal OR oral cavity OR laryngeal OR hypopharyngeal OR head OR neck) AND (cancer OR squamous cell carcinoma OR tumor OR neoplasm).” We also screened reference lists of the extracted articles to identify relevant ones.

### 2.2. Inclusion and Exclusion Criteria

Published research that satisfied the following inclusion criteria was included: (1) it explored the relationship between pretreatment CAR and long-term prognosis, including overall survival (OS), disease-free survival (DFS), or distant metastasis–free survival (DMFS) in HNC; (2) patients did not undergo any oncologic management such as operation or neoadjuvant therapy before CAR samples were obtained; and (3) the article reported necessary information for a meta-analysis. We excluded (1) studies that did not include survival outcomes and (2) letters, epidemiological studies, case reports, review articles, conference abstracts, and duplicate publications. Two authors (Chih-Wei Luan and Hsin-Yi Yang) independently reviewed the citations among the identified studies and enrolled studies that satisfied our inclusion criteria. The references were checked to identify potential eligible studies. A third author (Yao-Te Tsai) adjudicated on disagreements between the first two authors.

### 2.3. Data Extraction and Quality Assessment

The authors (Chih-Wei Luan and Hsin-Yi Yang) independently reviewed the included studies and extracted the following data using a standardized data collection form: study details (author(s), sample size, sex, publication year, study design (retrospective or prospective), and country of study), pathological characteristics (TNM staging), and clinical features (CAR cutoff values, treatment modality, survival outcome, and follow-up duration). The CRP/Alb ratio was calculated by dividing the serum CRP level by the serum albumin level with same scales as follows: CRP level (expressed in mg/L)/albumin level (expressed in g/L). A third author (Yao-Te Tsai) arbitrated when the first two authors disagreed.

### 2.4. Quality Assessment

The authors (Chih-Wei Luan and Hsin-Yi Yang) independently evaluated the quality of the included studies using the Newcastle–Ottawa quality assessment scale [17]. Studies with scores above 6 (maximum score of 9) were considered high quality. Again, disagreements between authors were resolved through discussion.

### 2.5. Statistical Analysis

The primary outcome measure of this study was the OS of patients with HNC. Secondary outcomes were DFS and DMFS. We pooled hazard ratios (HRs) with their corresponding 95% CIs to estimate the association between CAR and survival outcomes and determined heterogeneity using Cochran’s Q-test and the I^2^ statistic. All data pooling was performed using a random-effects model. Metaregression and cumulative meta-analysis by publication year, sample size, and cutoff values were employed to examine the potential influence of these factors on effect estimates. Subgroup analysis was performed to examine the significance of various potential moderators, including tumor sites, sample size, country of publication, cutoff values, and follow-up periods. To assess publication bias, Begg’s, Egger’s tests and a funnel plot were conducted. A *p* value of < 0.05 was defined as statistically significant in all calculations. Statistical analyses were conducted using STATA software version 15 (StataCorp LP, College Station, TX, USA).

## 3. Results

### 3.1. Study Characteristics

A structured search returned 110 records from PubMed, Embase, and CENTRAL. The flow diagram in Figure 1 shows the extraction process; 24 duplicate studies were excluded at first. Nineteen full-text articles were extracted for detail assessment after title and abstract screening. An additional eight articles were excluded in accordance with the inclusion and exclusion criteria. Finally, 11 studies involving 7080 patients with HNC were extracted for quantitative synthesis [18,19,20,21,22,23,24,25,26,27]. Table 1 reveals the characteristics of the studies extracted. In the extracted studies, sample sizes ranged from 40 to 2685 patients, and all studies were published between 2016 and 2020. Six studies focused on NPC, three studies were on oral cavity cancer, and an additional two studies investigated oropharynx and laryngeal cancers. The cutoff values for CARs varied from 0.03 to 0.525; the optimal CAR cutoff value was determined using Cutoff Finder in three studies [18,23,25] and a receiver operating characteristic curve analysis in seven studies [19,20,21,24,26,27,28]. The method for determining the cutoff values of CARs was not described in one study [22]. Further detail about include studies was summarized in Appendix A. Quality assessment showed that all included studies were of sufficient quality (Newcastle-Ottawa Scale, NOS scores: ≥7; Appendix A).

### 3.2. Prognostic Value of CAR for Predicting OS

Ten studies examined the value of pretreatment CAR for predicting OS; five, three, and two studies focused on NPC, oral cancer, and larynx and hypopharynx cancers, respectively. For the OS outcome, the pooled HR was 1.99 (95% CI: 1.57–2.53, *p* < 0.0001) for predicting mortality. The results suggest that a higher pretreatment CAR significantly predicts poor OS for patients with HNC (Figure 2).

### 3.3. Prognostic Value of CAR for Predicting DFS

Five studies (one on NPC, two on oral cancer, and two on larynx and hypopharynx cancers) provided HRs for the DFS of patients with HNC based on their CAR. Regarding DFS outcomes, the pooled results revealed that a CAR higher than the cutoff had an HR of 1.87 (95% CI = 1.28–2.75, *p* = 0.002) for predicting disease recurrence or mortality compared with CARs lower than the cutoff (Figure 2). These findings suggest that patients with HNC and a higher CAR were significantly associated with poor DFS rates and a higher treatment failure rate.

### 3.4. Prognostic Value of CAR in Predicting DMFS

Three reports that analyzed the data of 5013 patients estimated the influence of pretreatment CAR on DMFS. The pooled results indicated that a higher pretreatment CAR was significantly associated with a higher risk (HR = 0.46, 95% CI = 1.08–1.96, *p* = 0.001) of having a poor DMFS rate. This result implied that patients with HNC and a higher CAR might have a higher distant metastasis rate than those with a low CAR (Figure 2).

### 3.5. Metaregression

Metaregression by publication year or cutoff value revealed no statistical significances. However, sample size and effect estimates were significantly associated, regression coefficient = −0.0002268 (*p =* 0.008, Appendix A). A further cumulative analysis by sample size showed a progressive decline in HR with an increase in sample size (Appendix A).

### 3.6. Subgroup Analysis

We conducted a subgroup analysis to determine differences in the prognostic value of CARs for patients with HNC in various sites. When stratified by tumor location, CAR had a significant predictive value for OS in all subgroups, including those of patients with NPC, oral cancer, and hypolarynx and larynx cancers (Figure 3). CAR also had significant value for predicting DFS in all subgroups, including those of patients with NPC, oral cancer, and hypolarynx and larynx cancers (Figure 4). We also used subgroup analysis to further investigate potential moderators that could influence the HR of CAR in predicting OS (Table 2). A significant difference was observed between subgroup cutoff values and follow-up duration. However, differences among subgroups were identified in relation to tumor location (Figure 3 and Figure 4), sample size (Appendix A), and countries of origin (Appendix A). Despite the differences identified in relation to subgroups and various moderators, all subgroup pooling resultrevealed a significantly higher HR for patients with higher CARs above the cutoff value (Table 2).

### 3.7. Publication Bias

The funnel plot had apparent asymmetry on visual inspection, which demonstrates a lack of studies with small effect size HRs and small sample sizes for all three outcomes (Appendix A). When OS outcomes were focused on, the presence of publication bias was confirmed using Begg’s test (*p* = 0.009) and Egger’s tests (*p* < 0.001). Thus, we used the trim and fill method to evaluate the influence of publication bias on the pooled results. The results from the trim and fill calculation showed that the pooled HR was 1.57 (95% CI: 1.23–2.00, *p* < 0.001) for OS prediction after the missing studies were added. The new HR was smaller than that in the original results but did not change the direction and significance of these results. This finding supports the strength and stability of our meta-analysis.

## 4. Discussion

Despite previous meta-analyses providing evidence to support the prognostic value of CAR in relation to nasopharyngeal carcinoma, this is the first meta-analysis to demonstrate that CAR also has prognostic value with respect to other HNCs. Our results demonstrate that a higher pretreatment CAR is significantly associated with poor OS, DFS, and DMFS. The CAR holds prognostic prediction value for determining OS and DFS outcomes for all HNC tumor locations, including the mouth, nasopharynx, hypopharynx, and larynx (Table 2). Thus, CAR is a useful prognostic indicator for all HNCs.

The sample size of this study was a significant moderator in metaregression and subgroup analysis. These analyses showed that studies with small sample sizes tend to have higher HRs. This phenomenon can be explained by the presence of publication bias. Our funnel plot showed an obvious absence of studies with small sample size and small effect estimates. This finding suggests that publication bias might exist for a study with small size and that negative findings are less likely to be published, thus leading to higher HR values in studies with small sample sizes, as identified in metaregression and cumulative analysis.

In this work, tumor location and countries of origin were significant moderators in the subgroup meta-analysis. Given that only 11 studies were included in this meta-analysis, it was not appropriate to perform a multivariate metaregression to adjust for the potentially confounding influence of sample size on the moderating effect of tumor location and country of origin. However, oral cancer studies generally had smaller sample sizes than NPC studies (Figure 3). Furthermore, studies published by researchers in other countries usually had small sample sizes compared with studies by authors in China. Thus, we deduced that the moderating effects of tumor location and country of origin were confounded by the effect of sample size.

The underlying mechanism of the correlation between CAR and oncologic prognosis remains uncertain. Studies have indicated that inflammation participates in tumor development and progression by affecting the microenvironment of cancer pathogenesis. Systemic inflammatory responses may influence cancer cell proliferation, apoptosis, and angiogenesis, facilitating cell invasion and exacerbating cancer metastasis through the release of cytokines, immune cells, chemokines, acute phase proteins, and small inflammatory proteins [29,30,31]. Studies have provided supporting evidence that CARs are associated with cancer survival rates and serve as an independent prognostic marker [8,9,10,11,12,13]. CRP is synthesized in the liver and plays a role in acute inflammation, which is promoted by proinflammatory cytokine stimulation, such as interleukin-6 (IL-6) [32]. Additionally, hypoalbuminemia, as a chronic malnutrition indicator, was reported to be a biomarker of poor prognosis in patients with HNC [33]. When combined, elevated acute inflammation and decreased serum albumin may indicate nutritional deficiency, sarcopenia, and poor patient performance, which could all influence the prognosis of HNC [34]. Therefore, CAR is a valuable prognostic indicator and may provide additional prognostic information for individualized cancer management [35].

The TNM staging system, which is based on tumor factors, is always a crucial reference for treatment planning and prognostic prediction. However, in that system, host factors are not considered, such as nutritional status and systemic inflammation, which may account for disparate prognoses in patients with the same cancer stage. Therefore, despite similar TNM staging and advances in HNC therapy, equivalent treatment approaches may fail and lead to locoregional recurrent and distant metastasis.

Developing individualized treatment modalities with appropriate regimens is still difficult and often controversial. By assessing patients’ CAR, physicians can choose tailored treatment regimens for patients with HNC and inform their patients about management and long-term results [36,37]. Recently, Tsai et al. incorporated CAR in a multivariate prognostic nomogram for an oral squamous cell carcinoma analysis [28]. Their results suggested that CAR can help in prognosis prediction and inform therapy strategies. In addition, it is appropriate to add CAR to prognostic models of HNC. Thus, patients with a higher pretreatment CAR should be more closely monitored to prevent the poorer treatment outcomes observed in this subgroup as compared with patients with a lower pretreatment CAR. However, this relationship should be further tested in comprehensive clinical trials.

This meta-analysis had several limitations. First, included studies had considerable methodological diversity. Of the 11 included studies, tumor site, cancer stage, treatment method, CAR cutoff values, and follow-up periods all differed. Such diversity might have caused the observed statistical heterogeneity. Second, all included studies had exclusively Asian patient populations. Thus, it is difficult to generalize our findings to the global population. Third, the great variation in the cutoff value of CAR hinders its clinical application. Thus, future studies are needed to identify the exact cutoff of CAR to be clinically useful. Finally, this meta-analysis had obvious publication bias due to the lack of small studies with negative findings. However, our trim and fill analysis showed that publication bias did not alter the results of our meta-analysis. Nonetheless, we encourage the publication of small studies with negative findings.

## Figures and Tables

**Figure 1 diagnostics-11-00403-f001:**
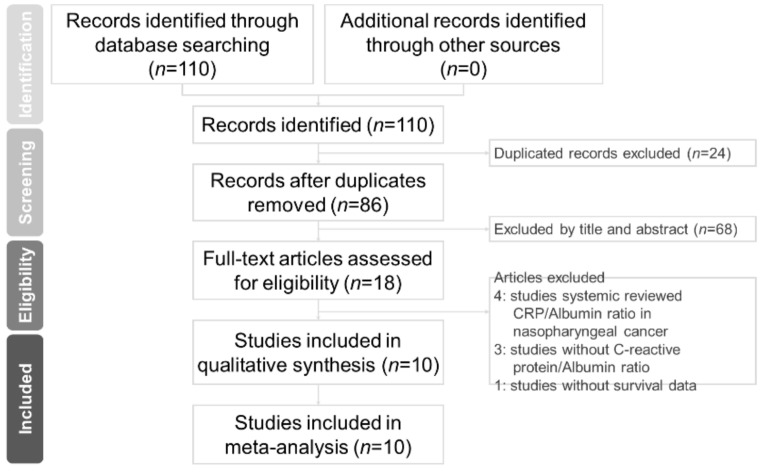
Flow diagram of study selection.

**Figure 2 diagnostics-11-00403-f002:**
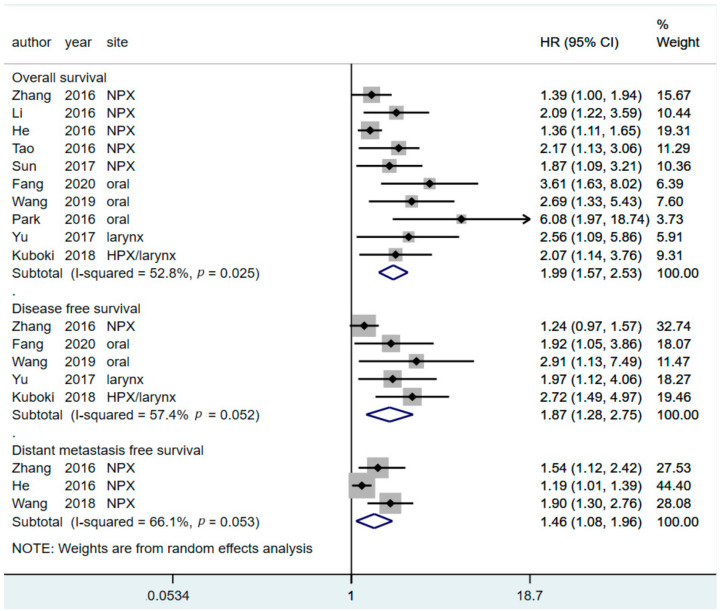
Forest plot showing the hazard ratios of high CARs for predicting overall survival, disease-free survival, and distant metastasis–free survival in patients with head and neck cancer. HR, hazard ratio; CI, confidence interval; CAR, C-reactive protein-to-albumin ratio.

**Figure 3 diagnostics-11-00403-f003:**
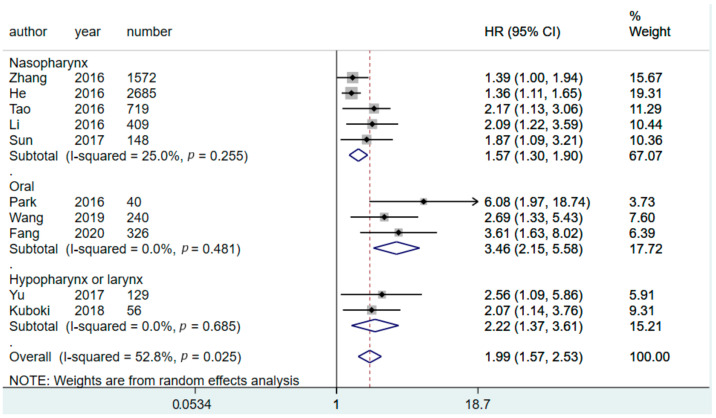
Forest plot depicting the pooled results of HRs for OS related to various tumor locations. HR, hazard ratio; CI, confidence interval; OS, overall survival.

**Figure 4 diagnostics-11-00403-f004:**
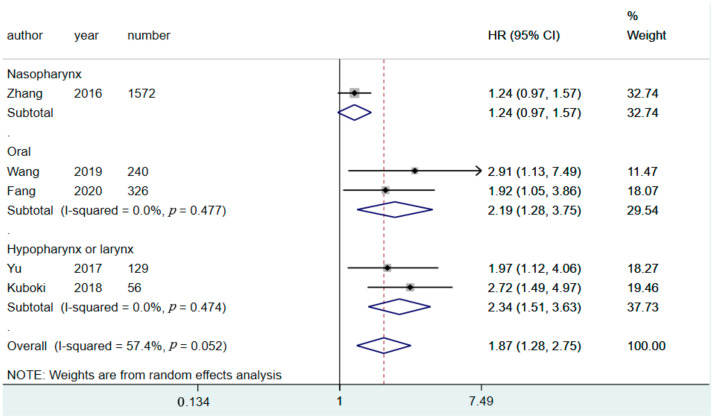
Forest plot showing the pooled results of HRs of DFS for various tumor locations. HR, hazard ratio; CI, confidence interval; DFS, disease-free survival.

**Table 1 diagnostics-11-00403-t001:** Demographic characteristics of patients in the included studies.

First Author	Year	Country	Sample Size	Tumor Site	Cancer Stage	Curative/ Palliative	Treatment	Cut-Off Value Resource	CAR Cut-Off (mg/g)	Outcome	Median Follow-Up (Months)	NOS
Li et al. [22]	2016	China	409	NPX	I–IV	Curative	R/C+R	Unknown	0.03	OS	53.7	7
He et al. [20]	2016	China	2685	NPX	I–III	Curative	R/C+R	ROC curve	0.064	OS, DMFS	46.3	7
Zhang et al. [27]	2016	China	1572	NPX	I–IV	Curative	R/C+R	ROC curve	0.05	OS	50	8
Wang et al. [18]	2018	China	756	NPX	I–IV	Curative	R/C+R	Cutoff Finder	0.081	DMFS	68.8	8
Tao et al. [24]	2016	China	719	NPX	I–IV	Curative	R/C+R	ROC curve	0.141	OS	47	7
Sun et al. [23]	2017	China	148	NPX	IVb	Palliative	C	Cutoff Finder	0.189	OS	21.8	8
Kuboki et al. [21]	2018	Japan	56	HPX/Larynx	I–IV	Curative	OP	ROC curve	0.32	OS, DFS	38	8
Yu et al. [26]	2017	China	129	Larynx	I–IV	Curative	OP/OP+C+R	ROC curve	0.047	OS	77	7
Wang et al. [25]	2019	China	240	Oral	I–IV	Curative	OP/OP+C+R	Cutoff Finder	0.525	OS, DFS	72.39	8
Park et al. [19]	2016	Korea	40	Oral	I–IV	Curative	OP/OP+C+R	ROC curve	0.085	OS, DFS	35.38	7
Fang et al. [28]	2020	Taiwan	326	Oral	I-IV	Curative	OP/OP+C+R	ROC curve	0.195	OS, DFS	48	7

NPX, nasopharynx; HPX, hypopharynx; C, chemotherapy; R, radiotherapy; CAR, C-reactive protein-to-albumin ratio; DFS, disease-free survival; DMFS, distant metastasis–free survival; NOS, Newcastle–Ottawa Scale score; OS, overall survival; OP, operation; ROC, receiver operating characteristic curve.

**Table 2 diagnostics-11-00403-t002:** Subgroup analysis.

Subgroup	Number of Studies	Number of Patients	Pooled HR with 95% CI	Subgroup *p* Value	Heterogeneity	
					I^2^ %	*p* Value
**Overall Survival**						
Tumor location						
Nasopharynx	5	5533	1.57 (1.30–1.90)	0.03	25	0.26
Hypopharynx/larynx	2	185	2.22 (1.37–3.61)		0	0.68
Oral cavity	3	606	3.46 (2.15–5.58)		0	0.48
Sample size						
≤500	7	5302	2.39 (1.86–3.09)	0.001	0	0.54
>500	3	1022	1.44 (1.22–1.69)		33	0.23
Country						
China	7	5902	1.57 (1.36–1.81)	0.01	35	0.16
Others	3	422	2.89 (1.86–4.48)		37	0.20
Cut-off value for CAR						
≤0.1	5	4706	1.77 (1.27–2.47)	0.26	60	0.04
>0.1	5	1618	2.27 (1.73–2.97)		0	0.71
Follow up period						
≥50 months	4	2350	1.88 (1.34–2.64)	0.62	33	0.21
<50 months	6	3974	2.14 (1.48–3.09)		65	0.01
**Disease Free Survival**						
Tumor location						
Nasopharynx	1	1572	1.24 (0.97–1.57)	0.02	0	NA
Hypopharynx/larynx	2	185	2.34 (1.51–3.63)		0	0.47
Oral cavity	2	566	2.19 (1.28–3.75)		0	0.48

CAR, c-reactive protein-to-albumin ratio; NA, not available.

## Data Availability

The data presented in this study are available on request from the corresponding author.

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
