# Peer review of "Prognostic Value of C-Reactive Protein-to-Albumin Ratio in Head and Neck Cancer: A Meta-Analysis"

_diagnostics, 2021, doi:10.3390/diagnostics11030403_

Round 1

Reviewer 1 Report

I liked the review, it was well structured, and it was done correctly from a methodological point of view. The article uses statistical approaches to compare the results obtained by different authors. Possible reasons for the discrepancies between the data of different authors are discussed. The approach itself is different from those used when writing reviews. That is why, despite the fact that the very value of the C-reactive protein-to-albumin ratio is a long-known prognostic factor, the article is of interest to readers. What is new is the statistical comparison and analysis of differences in the described literature data.

In paragraphs 218 to 233, the references to literature are incorrectly formatted. I believe that the article can be recommended for publication.

Author Response

Dear Reviewer:

We are happy to learn that our manuscript Prognostic value of C-reactive protein-to-albumin ratio in head and neck cancer: A meta-analysis had been reviewed by Diagnostics. We greatly appreciate your consideration and have amended the manuscript according to the reviewers’ suggestions. Our full responses to the reviewer’s comments are attached below, and the corresponding changes are shown in red in the revised manuscript.

Reviewer 1

I liked the review, it was well structured, and it was done correctly from a methodological point of view. The article uses statistical approaches to compare the results obtained by different authors. Possible reasons for the discrepancies between the data of different authors are discussed. The approach itself is different from those used when writing reviews. That is why, despite the fact that the very value of the C-reactive protein-to-albumin ratio is a long-known prognostic factor, the article is of interest to readers. What is new is the statistical comparison and analysis of differences in the described literature data.

Opinion 1: 

In paragraphs 218 to 233, the references to literature are incorrectly formatted. I believe that the article can be recommended for publication.

Reply:

This suggestion is very important, thank you for your reminder. We correct the reference format (line 218-233).

Thank you again for your time and consideration in reviewing this manuscript.

Reviewer 2 Report

The authors write about the prognostic value of the C-reacative protein to albumin ratio (CAR) in head and neck cancer patients. That’s actually very interesting, because it can be easily transferred into clinical routine. The paper is generally well written and the meta-analysis seems to be calculated correctly in many parts.

Nevertheless, I have some major concerns:

  • The authors state several times that CAR is a proved prognostic marker in nasopharyngeal carcinoma (NPC) and want to answer in this paper whether this is also true for all head and neck cancer patients. Unfortunately, 6 of the 11 studies and about 90 percent of the patients included are NPC. That culminates in results of section 3.4 where only NPC studies are included. This is methodically wrong, because effects of NPC will predominate by far. Key messages of the paper have to address this fact and it has to be confirmed what is already known for NPC and if it is true for the other sites of head and neck cancer.
  • The paper lacks about some basic information about CAR. Currently it does not provide information for the reader that can be transferred to clinical practice:
    1. How is CAR calculated? What units and scales are used? mg/L, g/L, mg/DL, g/DL?
    2. Have this units and scales be standardized?
    3. What is the normal range in healthy individuals?
    4. What have been the range, mean and median values in the included studies?
    5. The CAR cut off values in Table 1 differ by the power of ten. How can this be explained? Which cut off value should be used? Rolling the dice?
    6. May be a four-tier system (normal, low, moderate, strong elevated CAR) brings more light into this by using Kaplan-Meier-curves (accordingly to UICC stage)
    7. What was the patient number n=? Adding up the sample sizes in Table 1 does no result in 7080.
    8. What was the time point in calculating hazard ratio?
    9. Is CAR associated with tumor stage or is it an independent outcome parameter? From the clinical side it would be interesting to identify those patients who die earlier anyway because of general conditions such as elevated CAR. Maybe this is masked by other parameters in a long observation period.

Round 2

Reviewer 2 Report

May be I didn´t notice, but the tables with new content from the answers to the reviewers should be included, at least in the suppl material. After this no further objections from my side.

Author Response

Reviewer 2

May be I didn´t notice, but the tables with new content from the answers to the reviewers should be included, at least in the supplement material. After this no further objections from my side.

Response:

Thanks for your command. I add a table S2 in the supplement with all the detail of included studies. Thank you again for your time and consideration in reviewing this manuscript.

This manuscript is a resubmission of an earlier submission. The following is a list of the peer review reports and author responses from that submission.